# Toward Interpretable 3D Diffusion in Radiology: Token-Wise Attribution for Text-to-CT Synthesis

**Aidan Bradshaw**[1]                                    ABRADSHAW@ANDREW.CMU.EDU
**Katelyn Morrison**[1]                                   KCMORRIS@ANDREW.CMU.EDU
**Arpit Mathur**[1]                                        ARPITMAM@ANDREW.CMU.EDU
**Weicheng Dai**[2]                                               WD2119@BU.EDU
**Motahhare Eslami**[1]                                    MESLAMI@ANDREW.CMU.EDU
**Kayhan Batmanghelich**[2]                                       BATMAN@BU.EDU
**Adam Perer**[1]                                              ADAMPERER@CMU.EDU
[1] *Human-Computer Interaction Institute, Carnegie Mellon University, Pittsburgh, USA*
[2] *Department of Electrical and Computer Engineering, Boston University, Boston, USA*

**Editors:** Accepted for publication at MIDL 2025

## Abstract

Diffusion-based generative models have emerged as powerful tools for synthesizing anatomically realistic computed tomography (CT) scans from free-text prompts but remain opaque when delineating token influence on the conditioned CT volume. This lack of interpretability limits their clinical applicability, trustworthiness, and adoption across diagnostic and decision-support scenarios. We present a token-wise voxel attribution method for 3D text-to-image diffusion models that leverages cross-attention in U-Net–based architectures to extract individual token attention maps for synthetic CT scans. Our method visualizes individual, joint, or aggregated token-level voxel attributions during CT synthesis, helping to alleviate concerns about model transparency. This lays the groundwork for practical methods and structured explanations illustrating what aspects of attribution work well, where current limitations lie, and how researchers might approach explainable AI for 3D text-to-image diffusion models in radiology moving forward.

**Keywords:** 3D Medical Image Synthesis, Text-to-Image Diffusion, Explainable AI (XAI)

## 1. Introduction

Text-to-image diffusion models are being increasingly utilized in medical imaging for generating synthetic anatomies conditioned on text prompts (Han et al., 2024; Wilde et al., 2023). As these models scale to higher resolutions and generative reliability, they show promise for data augmentation (Zhang et al., 2023), training support (Bluethgen and Chambon, 2025), and clinical decision-making (Morís et al., 2024). However, clinical adoption remains limited by their black-box nature in high-dimension modalities like computed tomography.

To improve transparency, explainability techniques such as gradient-based saliency (Selvaraju et al., 2020), keyword heatmaps (Chefer et al., 2023; Evirgen et al., 2024), and attentive attribution (Tang et al., 2023) have been proposed. Still, many methods face three key limitations. First, few extend to 3D generative models, and the most performant methods (*i.e. gradient-based saliency*) are computationally laborious even on medium-end scientific computing resources. Second, model-intrinsic evaluation is emphasized over clinical usability (Reddy et al., 2021). Third, these methods produce static and dense, full-gradient

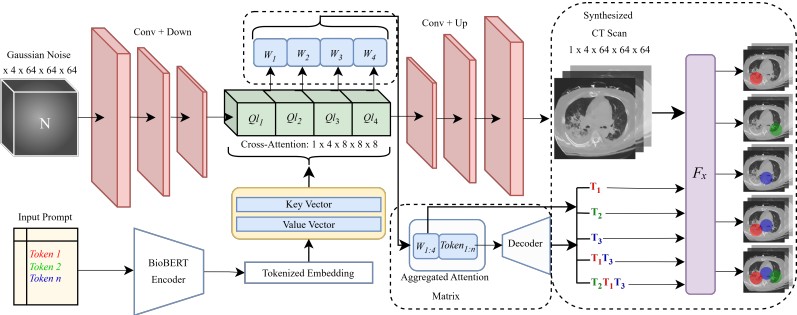

Figure 1: Overview of our token-wise attention and aggregation method

heatmaps that are difficult for clinicians and trainees to interpret and interact with (Herm, 2023).

We propose a lightweight, post hoc explainability method for 3D text-to-image diffusion models tailored to radiologists and trainees. Using a U-Net–based architecture, MedSyn (Xu et al., 2024), we extract token-wise cross-attention maps during inference and construct binary masked, word-attention volume pairs of individual and joint token attributions, captured in Figure 1. These token-wise attention volumes can be overlaid onto the synthesized CT to highlight spatial regions influenced by individual or combined prompt tokens (Figure 2). Our method provides an efficient, flexible, and interactive lens into model behavior, laying the groundwork for clinically meaningful explainability in 3D generative imaging.

## 2. Methods

### 2.1. Token-Wise Cross Attention Extraction

During inference, MedSyn downsamples the random Gaussian noise volume using denoising diffusion implicit models (DDIM) (Song et al., 2021), to a latent space, where it is cross-attended to using four bottleneck layers and text embeddings from BiomedVLP-CXR-BERT-specialized (Boecking et al., 2022). To compute token-level attention graphs, spatial features are flattened into queries $Q \in \mathbb{R}^{B \cdot F \times H \times Q_L \times D}$, where $B$ is the batch size, $F$ the number of slices, $H$ the number of heads, $Q_L$ the number of query positions per slice, and $D$ the head dimension. Token embeddings form keys and values $K, V \in \mathbb{R}^{B \cdot F \times H \times T \times D}$ for $T$ tokens. Cross-attention weights are computed as: $A^{(l,t)} = \text{softmax}\left(\frac{QK^\top}{\sqrt{D}}\right) \in \mathbb{R}^{B \cdot F \times H \times Q_L \times T}$ for layer $l \in \{1 : 4\}$ at timestep $t$.

**Attention Weight Aggregation.** At each of the $T_{\text{DDIM}} = 50$ timesteps, we record attention maps $A_t^{(l)}$ from the $L = 4$ mid-block CrossAttention layers. These are: (1) averaged over heads, $\bar{A}_t^{(l)} = \frac{1}{H} \sum_{h=1}^{H} A_{t,h}^{(l)}$; (2) concatenated across layers; and (3) aggregated over all timesteps to produce the final attention map: $\bar{A}_{\text{agg}} = \frac{1}{T_{\text{DDIM}} \cdot L} \sum_{t=1}^{T_{\text{DDIM}}} \sum_{l=1}^{L} \bar{A}_t^{(l)} \in \mathbb{R}^{B \cdot F \times Q_L \times T}$. This tensor is reshaped to $\mathbb{R}^{B \times F \times Q_L \times T}$ for downstream use.

**Token-Wise Attention Volumes.** To isolate per-token voxel attribution, we iterate over each token index $j \in \{1, \ldots, T\}$ and extract: $H_j = \bar{A}_{\text{agg}}[0, :, :, j] \in \mathbb{R}^{F \times Q_L}$ Each $H_j[f] \in \mathbb{R}^{Q_L}$ is reshaped into a $64 \times 64$ spatial map, and the $F = 64$ slices are stacked to form the volumetric saliency map $V_j \in \mathbb{R}^{64 \times 64 \times 64}$.

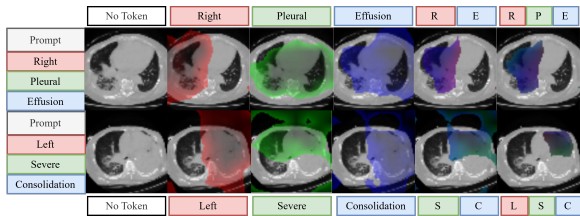

Figure 2: Axial slices showing token-level attribution overlays on generated CT scans.

## 2.2. Post Inference Voxel Aggregation

Given per-token saliency volumes $V_j \in \mathbb{R}^{64 \times 64 \times 64}$ for $j \in 1, \ldots, T$, we sparsify each volume by thresholding to retain only the top 25% most salient voxels. The threshold $\tau_j$ is defined as the 75$^{\text{th}}$ percentile of nonzero voxel-wise activations in $V_j$, yielding a binary mask $M_j = \mathbf{1}[V_j \geq \tau_j]$ that suppresses low-density confidence regions. Each mask is created based on a threshold and color-coded to preserve token salience. To model overlapping or interacting token influences, we define the filtered graph $\tilde{V}_j = M_j \odot V_j$, where $\odot$ denotes elementwise multiplication. When multiple tokens are selected (e.g., "severe" and "consolidation"), their filtered saliency maps $\{\tilde{V}_j\}_{j \in \mathcal{S}}$ are stacked and aggregated voxel-wise: $\tilde{V}_{\text{agg}}(x, y, z) = \frac{1}{|\mathcal{S}_{x,y,z}|} \sum_{j \in \mathcal{S}} \tilde{V}_j(x, y, z)$, where $\mathcal{S}_{x,y,z} = \{j \mid \tilde{V}_j(x, y, z) > 0\}$. Averaging is restricted to tokens with active influence at voxel $(x, y, z)$, avoiding dilution in jointly attended regions. The resulting volume $\tilde{V}_{\text{agg}} \in \mathbb{R}^{64 \times 64 \times 64}$ is overlaid on the CT scan (Fig. 2).

## 3. Conclusion and Future Directions

This work proposes a lightweight post hoc attribution method for 3D text-to-image diffusion models, enabling spatially grounded visualization of how individual text tokens influence generated CT volumes. This approach provides a push toward flexible mechanisms that probe model behavior and surface foundational challenges that we invite the community to consider for clinically relevant explainability in high-dimensional, generative models.

As the field continues exploring diffusion-intrinsic explainability methods, it is necessary to reconsider how to evaluate such methods in clinical contexts. Although previous approaches for 2D have been successful in running large-scale evaluations with radiologists or segmentation models, the 3D space requires attention to computational constraints under which these models will be deployed, the flexibility of interactive explanations in practical use, and the visual design of explanations themselves. Attention-based techniques show promise but must be interpreted cautiously. Word-to-CT visualizations should not be assumed as injective prompt-to-ground truth pathology segmentation but as noisy semantic artifacts, given different tokens and the stochastic process of DDIM sampling can yield similar attention patterns.

Our ongoing work aims to validate these attention maps across multiple diffusion-based architectures through radiologist-in-the-loop user studies. To support this, we are integrating this method into a custom-built OHIF (Hafey et al., 2019) viewer to assess usability and diagnostic value in clinical usage. We also plan to study how radiologists interpret overlays during diagnostic tasks to avoid misassumptions about model focus. We hope this work helps shape a broader, clinically grounded framework for interpretable 3D generative AI.

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
