# OpenReview forum: "Toward Interpretable 3D Diffusion in Radiology: Token-Wise Attribution for Text-to-CT Synthesis"
_MIDL.io/2025/Short_Papers — MIDL 2025 - Short Papers_

### Official Review · Reviewer_z1Dv · 2025-04-26

**Rating:** 3
**Confidence:** 4

**Summary:**

This paper introduces a novel token-wise voxel attribution method for 3D text-to-image diffusion models used in radiology, specifically focusing on Text-to-CT synthesis. The authors address the critical challenge of interpretability in these powerful but opaque generative models, which has hindered their clinical adoption despite their potential for data augmentation and decision support. Their approach extracts cross-attention maps from U-Net-based architectures to create visualizations showing how individual text tokens influence specific regions in generated CT volumes. By aggregating attention weights across multiple dimensions and applying thresholding techniques, they produce color-coded overlays that highlight areas of interest corresponding to specific prompt tokens. The method is designed to be computationally efficient compared to gradient-based alternatives and focuses on clinical usability rather than just model-intrinsic metrics.

**Strengths:**

- The work tackles the important challenge of interpretability in 3D medical image generation, which is crucial for clinical adoption.
- Unlike computationally expensive gradient-based methods, this attention-based approach is more efficient and practical.
- The method enables visualization of how specific words in the prompt influence different anatomical regions in the generated CT.

**Weaknesses:**

- The paper lacks quantitative evaluation or user studies with radiologists to validate that the visualizations accurately reflect meaningful relationships between text and image features.
- The method is implemented on a specific architecture (MedSyn), and its generalizability to other diffusion models isn't thoroughly demonstrated.

---

### Decision · Program_Chairs · 2025-05-01

Accept